# Investigation of Erase Cycling Induced Joint Dummy Cell Disturbance in Dual-Deck 3D NAND Flash Memory

**DOI:** 10.3390/mi14101916

**Published:** 2023-10-09

**Authors:** Kaikai You, Lei Jin, Jianquan Jia, Zongliang Huo

**Affiliations:** 1Institute of Microelectronics, Chinese Academy of Sciences, Beijing 100029, China; youkaikai20@mails.ucas.ac.cn (K.Y.); jiajianquan20@mails.ucas.ac.cn (J.J.); huozongliang@ime.ac.cn (Z.H.); 2University of Chinese Academy of Sciences, Beijing 100049, China

**Keywords:** dual-deck 3D NAND flash, joint-DMYs, erase cycling, electron backward injection

## Abstract

To satisfy the increasing demands for more word-line (WL) layers, the dual-deck even triple-deck architecture has emerged in 3D NAND Flash. However, the new reliability issues that occurred at the joint region of two decks became a severe challenge for developing multiple-deck technology. This work reported an abnormal reliability issue introduced by erasing disturbance of the dummy WLs at the joint region (Joint-DMYs) under multiple cycling. More specifically, after several erase cycling stresses, the increasing joint-DMY’s threshold voltage (Vt) due to the operational stress will finally result in additional disturbance to the adjacent data WLs. In this paper, we proposed this disturbance during erase originates from the backward injected electrons through FN tunneling based on our TCAD simulation result. Moreover, we also proposed an optimal erase scheme to reduce the backward electron injection and suppress the abnormal joint-DMY disturbance during the erase cycling.

## 1. Introduction

Three-dimensional NAND Flash has become a competitive nonvolatile storage technique because of its high-density and low-cost properties. In recent years, the number of word lines (WL) layers has continuously increased to fulfill the demand for scaling trend of the areal density. However, the increased number of WL layers leads to challenges to etching through the channel hole at once, even with state-of-the-art etching technology. Dual-deck architecture comes into the picture in 3D NAND technology to further achieve higher WL layers without requiring more sophisticated etching capability. Dual-deck architecture is composed of lower and upper decks. The number of WLs need to be etched in once have been shrunk since lower and upper decks can be produced from different etching processes [1,2,3,4,5]. Usually, a joint oxide is introduced between the upper and lower decks for better alignment. Meanwhile, WLs near the joint oxide are defined as joint-DMYs instead of normal storage WLs to suppress etching and alignment process-related reliability issues. From a reliability perspective, joint-DMYs will skip the program and erase the operation. However, joint-DMYs may still suffer disturbance during erase operation and further lead to adjacent WL interference effects [6,7,8]. In this work, the phenomenon of joint-DMYs Vt shift during erase cycling is reported, and it found that the backward electron injection during erase cycling is responsible for the joint-DMYs Vt shift. The larger channel diameter at the lower deck top WL can lead to more intensive backward electron injection, which is attributed to the higher electrical field perpendicular to the channel during the erase cycling. TCAD simulation has been carried out to interpret the experimental data. Furthermore, an optimal erase scheme has been proposed to replace the conventional ISPE, which can suppress cycling-induced joint-DMYs Vt shift, and has been validated by experiments.

## 2. Materials and Methods

In this work, the experiment is based on charge-trapping vertical channel 3D NAND flash test chips, with dual-deck architecture. Figure 1a shows the structure of the dual-deck 3D NAND array, the Figure 1b is the TCAD simulation of joint-DMYs and data WLs. In addition, the Sentaurus TCAD simulator is used for TCAD simulation. In the charge trapping layer, the drift-diffusion model and charge trapping/de-trapping model are taken into account. Meanwhile, the non-local tunneling (NLT) model through the tunneling layer and blocking layer is considered. The interaction between free carriers and trapped carriers is governed by the carrier capture phenomenon calculated by Shockley-Read-Hall (SRH) theory and carrier emission contributed by thermal and Poole-Frenkel effect. Dual-deck channel was constructed in TCAD simulation, and there are two joint-DMYs between the lower and upper decks. As shown in Figure 1b, the joint-DMY at the bottom of upper-deck is DMY1 and the joint-DMY at the top of lower-deck is DMY0. In order to simplify the simulation, we just chose the 64 WL layer 3D NAND structure to construct the TCAD simulation. Table 1 is the detailed device structure parameters of the simulation, and the simulation setting is almost the same as our previous work [9,10,11,12,13]. Figure 2 is the DMY0 IdVg curve before erasing. The TCAD simulation data used with Table 1 parameter is fitted with the experiment data. Actually, the experiment has a GIDL (gate-induced drain leakage) current if the Vg < −2 V, because of GIDL current has no impact on this study, this related model and parameter are not included in the simplified TCAD simulation.

## 3. Experiments and Simulations

In order to characterize the Vt shift of DMY0 and DMY1 during erase cycling, the bias waveform diagram was shown as Figure 3a. And the test flow is firstly collect DMY0 and DMY1 initial Vt distribution, then programing all data WLs to random data pattern, finally incremental BL/SL erase pulse voltage (ISPE) is applied to implement full block erase and collect DMY0 and DMY1 disturbed Vt distribution. To ensure that DMY0 and DMY1 has consistent reliability performance, the DMY0 and DMY1 skipped program or erase operation. As shown in Figure 3b,c, with increasing erase pulse count, the distribution of the DMY0/1 Vt shifts towards higher Vt. The Vt distribution of the DMY0 at lower deck top position shifts higher than that of the DMY1 (at upper deck bottom position) after two erase pulses. Due to interference impact, the higher DMY0 Vt may impact the nearby WL Vt distribution such as WLn, which lead to WLn Vt disturbance [14,15,16,17].

For the physical mechanism, TCAD simulation was used to explain the backward injection during the erase operation. To simplify the TCAD simulation, only one erase pulse with high Vers (25 V) and Vdmy (20 V) is applied in the TCAD simulation, instead of incremental and multiple erase pulses. Before erase, the TCAD simulation structure is initial state which means the trap layer does not contain any electron/hole trapped charge. Figure 4a,b are the TCAD simulations of DMY0/1 IdVg curve, respectively. Before erase, DMY0 and DMY1 have almost the same Vth about −0.1 V. But after erase, DMY0 Vth is 3.79 V and DMY1 is 0.68 V. The Vth shift trend is consistent with experiment data in Figure 3b,c. According to the Figure 3 and Figure 4, the DMY0/1 have different Vt shift trend after erase cycling. DMY0 has lager channel hole diameter, so that lead to higher Vt shift than DMY1 after erase operation. The corresponding electron and hole trapped charge distribution is simulated as shown in Figure 5a,b The electron and hole trapped density at the dotted line is shown as Figure 5c and d, respectively. Between the DMY1 and the WLn+1, there are more hole trapped charge but less electron trapped charge than the DMY0 and WLn. It is due to that smaller channel diameter generates larger electric field vertical to poly channel direction which can lead to larger hole FN tunneling from the channel to trapping layer. In contrast, along with the poly channel direction, there is smaller electric field which lead to smaller electron FN tunneling from the WLn+1 metal gate to the trapping layer.

Figure 6 indicates the schematics of electrons and holes injection process to the trapping layer, and it explains the physical model of backward FN tunneling in detail. In the figure, the blue arrow represents the direction of the electric field between WL and joint DMYs, and this electric field will lead to the electron back side injected into the trap layer. Meanwhile, the red arrow represents the electric field between poly and WL, and it will lead hole injected into the trap layer during erase. The red and blue region in the trap layer represent hole and electron trap density, respectively. The hole injected into the trapping layer will lead to WLn and WLn−1 Vt shift down, but the electron backward injection into the trapping layer will accumulate during each erase pulse and will finally result in DMY0 Vt shift to higher.

Furthermore, samples with different block oxide layer thicknesses have been tested to validate the backward injection mechanism. The thicker blocking oxide theoretically suffers less electron backward injection due to the lower electric field perpendicular to the channel. Figure 7a,b show DMY0/1 Vt shift after the ISPE cycle under different block oxide thicknesses, It is obvious that thicker block oxide samples exhibit less DMY Vt shift, which means there are fewer electrons backward injected into the trapping layer through the blocking layer during erase. To resolve the ISPE cycling induced joint-DMYs Vt disturbance, a novel erase scheme is proposed to suppress the backward electron tunneling. Figure 8a is the diagram of the conventional and proposed erase scheme. The proposed erase scheme is with only one erase pulse. In a conventional erase scheme, joint-DMYs bias is increased in each ISPE pulse, while one erase pulse has a large erase time to guarantee enough erase efficiency. Considering the backward electron tunneling during erase, the proposed erase scheme avoids high voltage on the joint-DMYs, therefore it can suppress backward FN tunneling. To verify the improvement in joint-DMYs Vt shift of the proposed erase schemes, Figure 8b,c show array data comparisons of two erase schemes in the erase cycle. It is obvious that the proposed erase scheme almost has no DMY0 and DMY1 Vt shift phenomena in each erase cycle, while DMY0 Vt shows a significant shift after 3 erase cycles in the conventional condition.

In summary, the above experimental results validate that the proposed erase scheme can effectively suppress the backward electron injection during the erase operation. Therefore, decreasing the erase bias and one longer erase pulse width can prevent joint-DMY from cycling induced Vt shift.

## 4. Conclusions

To mitigate the joint-DMYs worse reliability issue during erase cycling, in this work, the joint-DMYs cell disturbance issue after erase cycling has been investigated in a dual-deck 3D NAND technology. From the analysis results, the Vt shift is attributed to the electron backward injection into the joint-DMYs cell trapping layer through the nearby WL blocking oxide. Meanwhile, the backward injection efficiency is related to channel hole diameter and block oxide thickness. A larger channel diameter and thicker block oxide have a higher backward electric field between joint-DMYs and nearby WL, which leads to more intensive backward injection. Finally, the proposed one pulse erase scheme, which has lower erase bias and lower electric field between joint-DMYs and nearby WL has been validated to suppress the joint-DMYs Vt shift.

## Figures and Tables

**Figure 1 micromachines-14-01916-f001:**
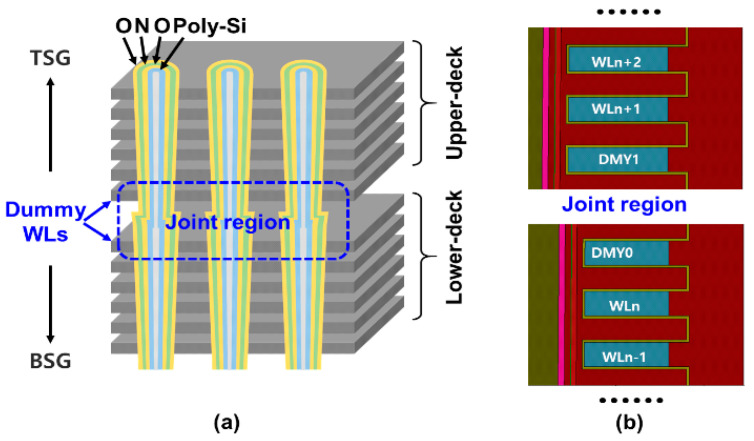
(**a**) The structure of dual-deck 3D NAND flash memory. (**b**) TCAD simulation structure.

**Figure 2 micromachines-14-01916-f002:**
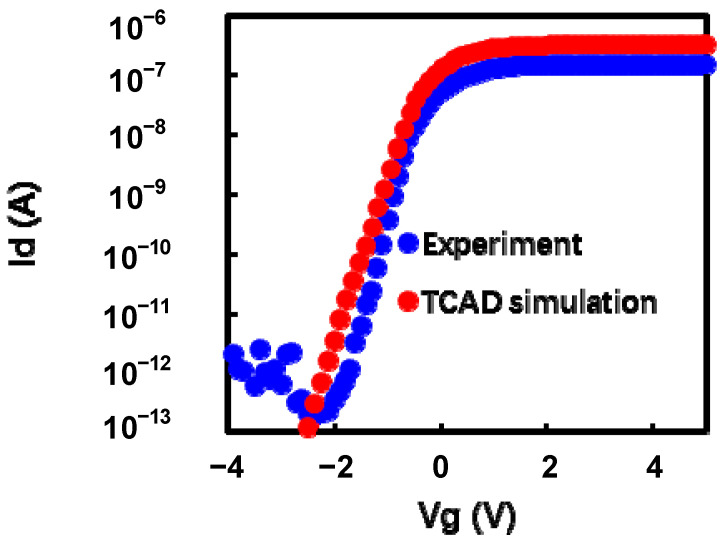
The comparison of experiment and TCAD simulation DMY0 IdVg curve before erase.

**Figure 3 micromachines-14-01916-f003:**
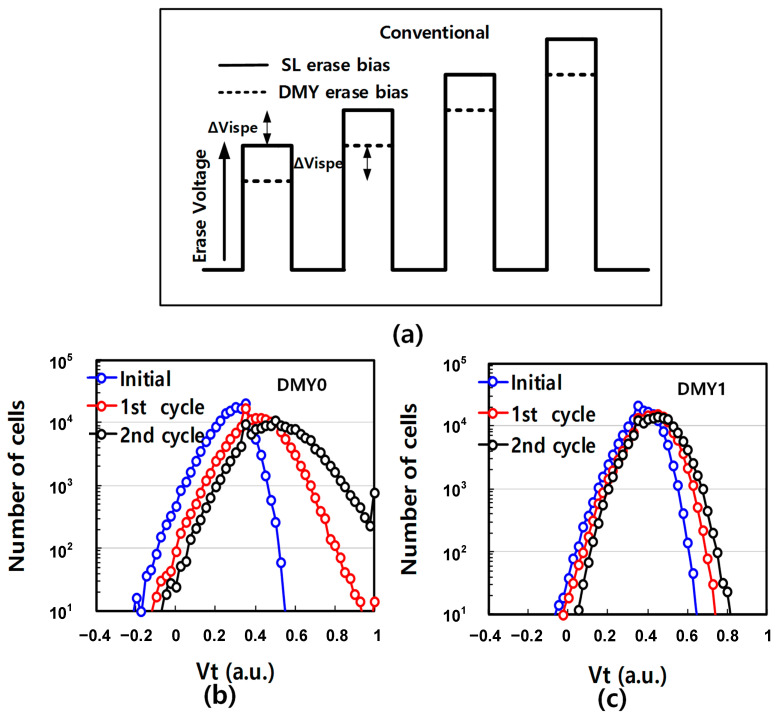
(**a**) The conventional timing diagrams of the erase operation. (**b**) Array level of Vt distribution at different erase pulse count of DMY0 and (**c**) DMY1.

**Figure 4 micromachines-14-01916-f004:**
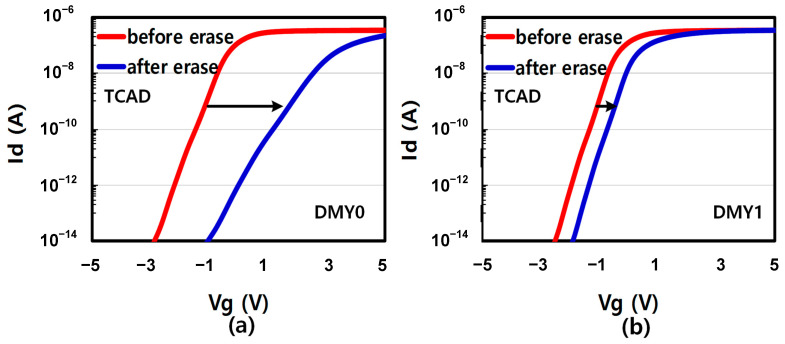
(**a**) TCAD simulated data of DMY0 before and after erase IdVg curve. (**b**) TCAD simulated data of DMY1 before and after erase IdVg curve.

**Figure 5 micromachines-14-01916-f005:**
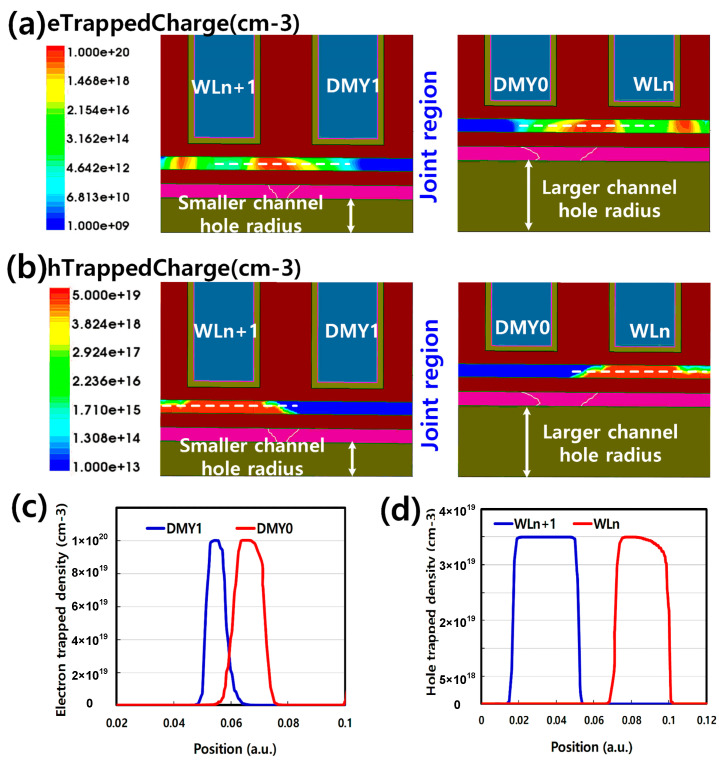
(**a**) The distribution of electron trapped charge in the trapping layer after erase. (**b**) The distribution of hole trapped charge in the trapping layer after erase. (**c**) DMY0/1 electron trapped charge density in the trapping layer dotted line after erase. (**d**) WLn/n+1 hole trapped charge density in the trapping layer dotted line after erase.

**Figure 6 micromachines-14-01916-f006:**
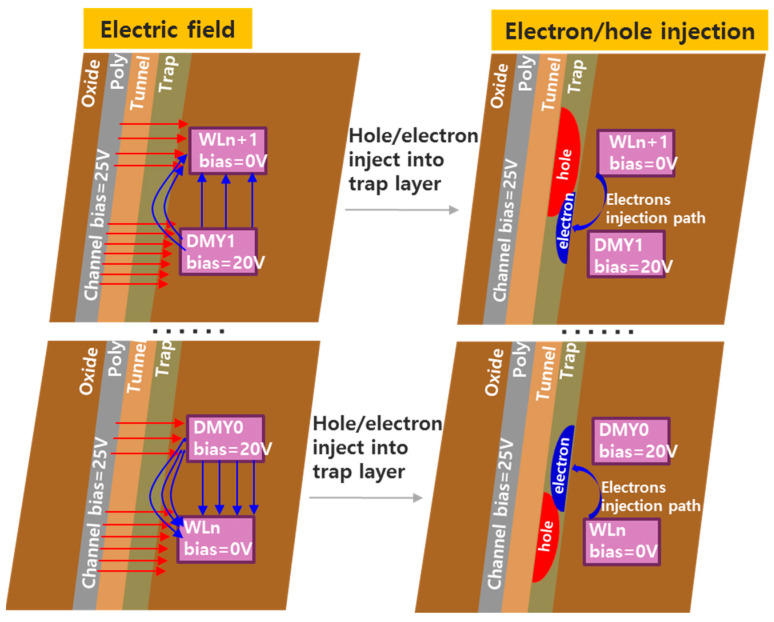
Schematics of electric field and hole/electron FN tunneling during erase.

**Figure 7 micromachines-14-01916-f007:**
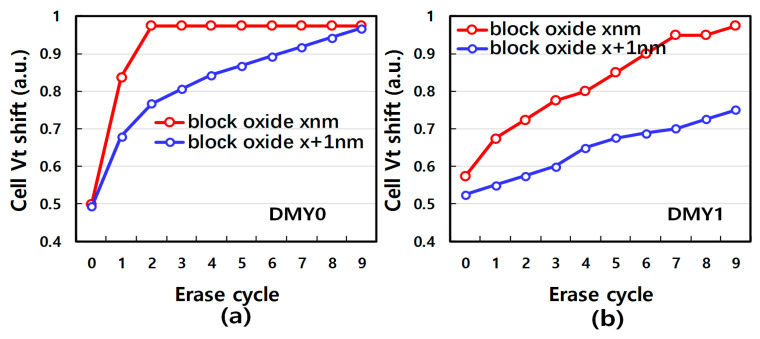
(**a**) Different block layer thickness array level of Vt median value under at different erase cycle count of DMY0 (**b**) DMY1.

**Figure 8 micromachines-14-01916-f008:**
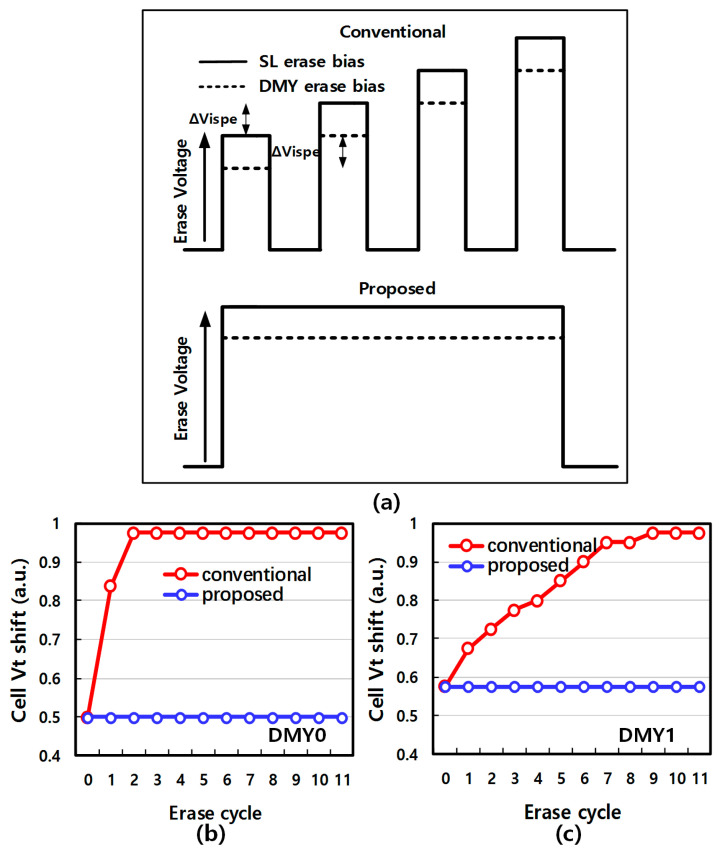
(**a**) The diagram of the conventional and proposed erase scheme. (**b**,**c**) show array data comparisons of two erase schemes in the erase cycle.

**Table 1 micromachines-14-01916-t001:** Main parameters used in the simulation.

Parameter	Value
SiO_2_ relative dielectric constant	3.9
SiO_2_ electron affinity	0.9 eV
SiO_2_ band gap	8.9 eV
SiO_2_ electron effective mass	0.39 m_o_
SiO_2_ electron effective mass	0.47 m_o_
Si_3_N_4_ relative dielectric constant	7.5
Si_3_N_4_ electron affinity	1.9 eV
Si_3_N_4_ band gap	5 eV
Si_3_N_4_ electron trap energy (from mid band gap)	1.2 eV
Si_3_N_4_ electron cross section	1 × 10^−20^ cm^2^
Si_3_N_4_ electron trap density	3 × 10^19^ cm^−3^
Si_3_N_4_ electron trap volume	1 × 10^−10^ um^−3^
Si_3_N_4_ relative dielectric constant	1 × 10^−10^ um^−3^
Si_3_N_4_ hole trap energy (from mid band gap)	−1.35 eV
Si_3_N_4_ hole cross section	1 × 10^−17^ cm^2^
Si_3_N_4_ electron trap density	1.35 × 10^17^ cm^−3^
Si_3_N_4_ electron trap volume	1 × 10^−10^ um^−3^
Poly-Si/SiO_2_ interface donor/accepter trap density (peak at valance band)	5 × 10^13^ cm^2^
Poly-Si/SiO_2_ interface donor/accepter trap Gaussian distribution sigma	0.05 eV

## Data Availability

Data is unavailable due to privacy.

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
