# Peer review of "Investigation of Erase Cycling Induced Joint Dummy Cell Disturbance in Dual-Deck 3D NAND Flash Memory"

_micromachines, 2023, doi:10.3390/mi14101916_

Round 1
Reviewer 1 Report
The author addressed the dummy word-lines effect on erase operation in dual deck VNAND. There are some points which you can improves the manuscript of TCAD simulation.
1. What is highlight novelty of this paper, compared with "https://www.mdpi.com/2079-9292/11/17/2738"?
2. TCAD simulation paper should show the calibration data for simulation accuracy.
3. Depending on the trap property of Nitride and tunnel oxide quality, the simulation data and tendency will be changed. If the calibration is not possible, the author should shows various trap conditions.
Author Response
Comments 1: What is highlight novelty of this paper, compared with "https://www.mdpi.com/2079-9292/11/17/2738"?
Our response:
Thank you for your comment. The innovative point of this paper is that this paper focus on investigate the dual-deck structure joint-DMYs Vt shift under erase cycling. In the future, dual deck even triple deck is the mainstream of 3D NAND. Therefore, the joint-DMYs is a crucial factor of 3D NAND reliability. The Beomsu’s paper seems not to mention this.
Comments 2: TCAD simulation paper should show the calibration data for simulation accuracy.
Our response:
Thank you for your comment. I have add the calibration data in my paper. After calibration, the TCAD simulation IdVg curve is match well with the experiment data.
Comments 3: Depending on the trap property of Nitride and tunnel oxide quality, the simulation data and tendency will be changed. If the calibration is not possible, the author should shows various trap conditions.
Our response:
Thank you for your comment. Yes, it will be change the simulation data and the tendency if we use different parameter in simulation, so we would calibrate the parameter and ensure the accuracy of TCAD simulation. Table 1 is our trap conditions, and we add this table in our paper.
Table 1.Main parameters used in the simulation.
|
Parameter |
Value |
|
SiO2 relative dielectric constant |
3.9 |
|
SiO2 electron affinity |
0.9eV |
|
SiO2 band gap |
8.9eV |
|
SiO2 electron effective mass |
0.39mo |
|
SiO2 electron effective mass |
0.47mo |
|
Si3N4 relative dielectric constant |
7.5 |
|
Si3N4 electron affinity |
1.9eV |
|
Si3N4 band gap |
5eV |
|
Si3N4 electron trap energy (from mid band gap) |
1.2eV |
|
Si3N4 electron cross section |
1e-20cm2 |
|
Si3N4 electron trap density |
3e19cm-3 |
|
Si3N4 electron trap volume |
1e-10um-3 |
|
Si3N4 relative dielectric constant |
1e-10um-3 |
|
Si3N4 hole trap energy (from mid band gap) |
-1.35eV |
|
Si3N4 hole cross section |
1e-17cm2 |
|
Si3N4 electron trap density |
1.35e19cm-3 |
|
Si3N4 electron trap volume |
1e-10um-3 |
|
Poly-Si/SiO2 interface donor/accepter trap density (peak at valance band) |
5e13cm2 |
|
Poly-Si/SiO2 interface donor/accepter trap Gaussian distribution sigma |
0.05eV |

Reviewer 2 Report
This paper starts off with a nice premise to understand the erase reliability in stacked word lines. However, it is all theoretical and we don't have any idea how this relates to real devices and thresholds. The English language is fairly difficult to understand what exactly the authors did, other than to say that they made a TCAD simulation. I think the physics should be better presented and explained how the read-write cycles leads to a change in threshold voltage shift. It is nice that you bring up an improvement but that seems counterintuitive that a continuous voltage rather than pulses will be more reliable.
There are many cases where it just reads improperly. I can't summarize them all here. Also, the paragraphs are not organized in a logical way.
Author Response
Comments 1: This paper starts off with a nice premise to understand the erase reliability in stacked word lines. However, it is all theoretical and we don't have any idea how this relates to real devices and thresholds. The English language is fairly difficult to understand what exactly the authors did, other than to say that they made a TCAD simulation. I think the physics should be better presented and explained how the read-write cycles leads to a change in threshold voltage shift. It is nice that you bring up an improvement but that seems counterintuitive that a continuous voltage rather than pulses will be more reliable.
Response 1: Thank you for your comments. We have revised our English language and add the comparison of simulation data and experiment data. And this work is based on our previous study ref. [1], this paper studied dual-deck and has introduced the threshold voltage shift during program operation in detail. But our previous work is focus on program operation, this work is focus on erase operation.
Reference:
- Jia et al., "A Novel Program Scheme to Optimize Program Disturbance in Dual-Deck 3D NAND Flash Memory," in IEEE Electron Device Letters, vol. 43, no. 7, pp. 1033-1036, July 2022, doi: 10.1109/LED.2022.3178155.

Round 2
Reviewer 2 Report
Thank you for your clarification. I see how the experiment fits the theory.